# Identification of tauopathy-associated lipid signatures in Alzheimer's disease mouse brain using label-free chemical imaging

Hao Meng[1], Alicia Elliott[1], Jessica Mansfield[1], Michelle Bailey[1], Mark Frogley [2], Gianfelice Cinque[2], Julian Moger[1], Nick Stone [1], Francesco Tamagnini [3,4] & Francesca Palombo [1] ✉

There is cumulative evidence that lipid metabolism plays a key role in the pathogenesis of various neurodegenerative disorders including Alzheimer's disease (AD). Visualising lipid content in a non-destructive label-free manner can aid in elucidating the AD phenotypes towards a better understanding of the disease. In this study, we combined multiple optical molecular-specific methods, Fourier transform infrared (FTIR) spectroscopic imaging, synchrotron radiation-infrared (SR-IR) microscopy, Raman and stimulated Raman scattering (SRS) microscopy, and optical-photothermal infrared (O-PTIR) microscopy with multivariate data analysis, to investigate the biochemistry of brain hippocampus in situ using a mouse model of tauopathy (rTg4510). We observed a significant difference in the morphology and lipid content between transgenic (TG) and wild type (WT) samples. Immunohistochemical staining revealed some degree of microglia co-localisation with elevated lipids in the brain. These results provide new evidence of tauopathy-related dysfunction in a preclinical study at a subcellular level.

Worldwide, there are currently more than 55 million people affected by dementia, a number which is predicted to double every 20 years. Approximately 60–70% of dementia cases are associated with Alzheimer's disease (AD)[1]. However, there is still a lack of definitive diagnostic tests for AD to date. AD is a neurodegenerative disease characterised by progressive cognitive dysfunction and memory loss. Patients with AD experience a set of biological and pathological changes in their brain, with the two most common hallmarks being amyloid-beta plaques and neurofibrillary tangles composed of abnormally accumulated amyloid-beta peptides and hyper-phosphorylated tau proteins, respectively[2]. In recent years, there has been growing evidence that lipid metabolic disorders are also intimately involved in AD development, potentially serving as a transition stage to the onset of hallmark features. Despite the lack of definitive evidence of the direct correlation between AD hallmarks with its risk factors, such as ageing, diabetes, obesity, and head injury, it is clear that lipid metabolism is highly susceptible to these factors, implying that the study of lipid metabolism might be crucial to shed light on AD pathophysiology[3]. Impaired lipid metabolism could lead to abnormal deposition of certain lipids in the brain during the ageing

process and in neurodegenerative diseases. Glial cells, such as microglia and astrocytes, are believed to play a crucial protective role in the maintenance of lipid homeostasis by storing lipids in droplets[4–6]. A remarkable accumulation of lipid droplets (LDs) and abnormal lipid levels have been observed in the inflammatory state of microglia caused by ageing and neurodegeneration[4]. The presence of elevated lipids associated with activated astroglia has been revealed in an experimental model of amyloidopathy[7]. The tau dysfunction in AD is known as AD tauopathy, characterised by the abnormal deposits of tau aggregates in the brain (mainly in neurons, also in glial cells and extracellular matrix). A recent study shows that LD accumulation impairs tau phagocytosis of microglia, thereby exacerbating tau aggregation and neuroinflammation in mouse models of tauopathy[8]. Despite the advances, the comprehension of the interplay between altered lipid content, activation of glial cells and AD hallmarks has so far remained limited.

Label-free imaging methods that use vibrational spectroscopy, such as those based on infrared absorption and Raman scattering, enable non-destructive all-optical screening of tissues, where molecular-specific spectral

[1]Department of Physics and Astronomy, University of Exeter, Exeter, EX4 4QL, UK. [2]Diamond Light Source, MIRIAM beamline B22, Harwell Science & Innovation Campus, Didcot, OX11 0DE, UK. [3]School of Pharmacy, University of Reading, Reading, RG6 6UB, UK. [4]Centro Studi Biomedici, Università degli Studi della Repubblica di San Marino, Salita alla Rocca, 44 – 47890 San Marino Città, Republic of San Marino. ✉e-mail: F.Palombo@exeter.ac.uk

and spatial features are probed with sub-cellular resolution[9–14]. In particular, Fourier transform infrared (FTIR) spectroscopic imaging and synchrotron radiation infrared (SR-IR) microscopy have largely been explored in the studies of amyloid-beta plaques in AD brains, showing the molecular profile of plaques in terms of a lipid-rich halo and beta-sheet dense core[7,9,15]. In addition, Raman microscopy has provided the molecular composition of finer structures in biological tissues in many studies[7,10,12,13,16]. With the aid of machine-learning methods such as principal components analysis (PCA) and k-means cluster analysis, some studies using Raman microscopy have achieved a chemical-specific and morphological elucidation of the amyloid plaques and surrounding structures[7,10,16]. Moreover, stimulated Raman (SRS) imaging has been more commonly employed in AD studies to characterize tissue samples, due to its higher sensitivity and lower fluorescence background[13,17–19]. Recently, Klementieva et al. have presented images of polymorphic structures of aggregated amyloid peptides in dendrites of AD transgenic neurons using an emerging super-resolution IR technique, optical-photothermal infrared (O-PTIR) microscopy[20]. Different from traditional IR techniques, O-PTIR enables IR imaging with submicrometric spatial resolution based on the photothermal infrared effect[21–24]. There have been many spectroscopic studies focused on amyloid plaques; however, to date, much less progress has been made in chemically detecting signatures associated with tauopathy[25,26], the other major hallmark of AD.

In this work, we explored the capability of complementary spectroscopic modalities, FTIR spectroscopic imaging, SR-IR, Raman, SRS and O-PTIR microscopy, for the study of a mouse model of AD-related tau pathology. Hyperspectral imaging was performed on ex vivo sections of both transgenic (TG) and wild type (WT) mouse hippocampus to detect chemical signatures of tauopathy. We also conducted immunohistochemical staining of the tissues to validate the results of the spectroscopic analysis. The morphological and chemical features of the mouse brain affected by AD-related tau pathology were revealed with enhanced spatial resolution in a label-free manner.

## Results

A schematic workflow of the experiments conducted in this work is illustrated in Fig. 1. Samples of brain hippocampus (which contains the early signatures of AD) from both TG and WT mice were analysed ex vivo in both dry and wet forms. It has been reported that the pyramidal layer and surrounding tissue in the *Cornu Ammonis* (CA1) region of the hippocampus show the most distinctive pathological features in AD progression[27,28]. And

so, the use of different vibrational spectroscopic methods was directed to the CA1 region in the order of increasing spatial resolution to detect fine structures within the tissue.

In the first part of the measurements, FTIR spectroscopic imaging was performed on dry sections to enable whole tissue analysis, followed by SR-IR, Raman and O-PTIR microscopy analysis of specific regions of interest (ROIs) within CA1. In the second part, SRS microscopy was applied to wet tissue sections across multiple ROIs, and finally validation was conducted with immunohistochemical staining and fluorescence imaging.

### FTIR, SR-IR, Raman and O-PTIR imaging

In Fig. 2a, we show an optical image of a TG mouse hippocampal section, where the darker curves represent the *Dentate Gyrus* and the *Cornu Ammonis*, with its sub-regions (CA1-CA3). (Some tissue damage is apparent, due to the sample preparation).

A micro-FTIR spectroscopic image of the tissue section based on the lipid ester signal (*C=O* stretching at 1761–1720 cm$^{-1}$) is presented in Fig. 2b. The *Dentate Gyrus* and pyramidal layer are highlighted in blue due to a lower lipid content compared with the surrounding tissue, which can be explained by the fact that neuronal bodies have relatively less lipids than the rest of the tissue.

Figure 2c, d show two SR-IR maps, one based on the Amide I (1716–1600 cm$^{-1}$) and the other one on the lipid ester band (1761–1720 cm$^{-1}$) from an ROI indicated by the black dashed box in Fig. 2a. In spectroscopic studies of biological tissues, the Amide I band is of great interest since it reflects the content and secondary structure of proteins. Here, the two maps are complementary in that the protein-rich structures in the pyramidal neurons are identified by red regions in Fig. 2c, whilst the pyramidal layer is showing a low-lipid intensity (blue stripe) in Fig. 2d.

Similar characteristics of the tissue are identified from the principal component analysis applied to a Raman map (Fig. 2e, f) from the region denoted by a red dashed box in Fig. 2a. The principal component 4 (PC4; Fig. 2e) depicts areas which are rich in both proteins and lipids, with peaks at 1659 cm$^{-1}$ (Amide I), 1446 cm$^{-1}$ (*CH$_2$* bending), 1296 cm$^{-1}$, 1127 cm$^{-1}$ (*C−N* stretching), and 1003 cm$^{-1}$ (*C−C* symmetric stretching). On the other hand, principal component 5 (PC5; Fig. 2f) directly reveals the lipid distribution in the tissue, with peaks at 1438 and 1296 cm$^{-1}$ (*CH$_2$* deformation), 1129 and 1064 cm$^{-1}$ (*C−C* skeletal stretching). All peak assignments are listed in Table 1.

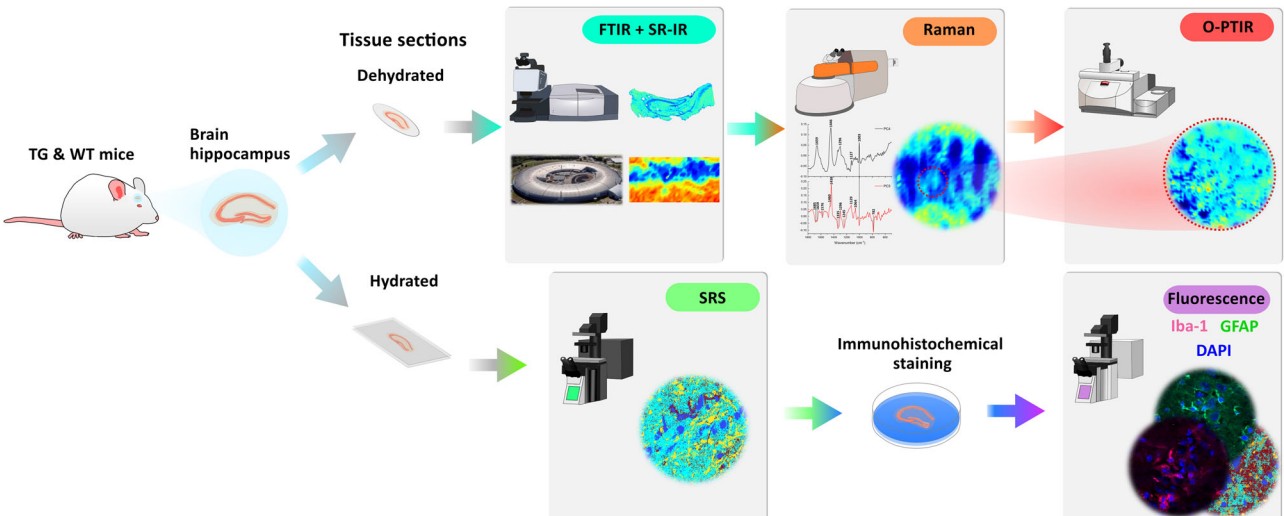

**Fig. 1 | Experimental workflow.** Visual illustration of our approach where a suite of micro-spectroscopic methods was applied following the order of increasing spatial resolution and decreasing field of view, from whole tissue to subcellular imaging. The brain hippocampus from TG and WT mice was sectioned and mounted on calcium fluoride slides, before being analysed by FTIR imaging, SR-IR, Raman and O-PTIR microscopy (top panel). Hydrated tissue sections were imaged using SRS microscopy followed by immunohistochemical staining and fluorescence imaging for validation (bottom panel).

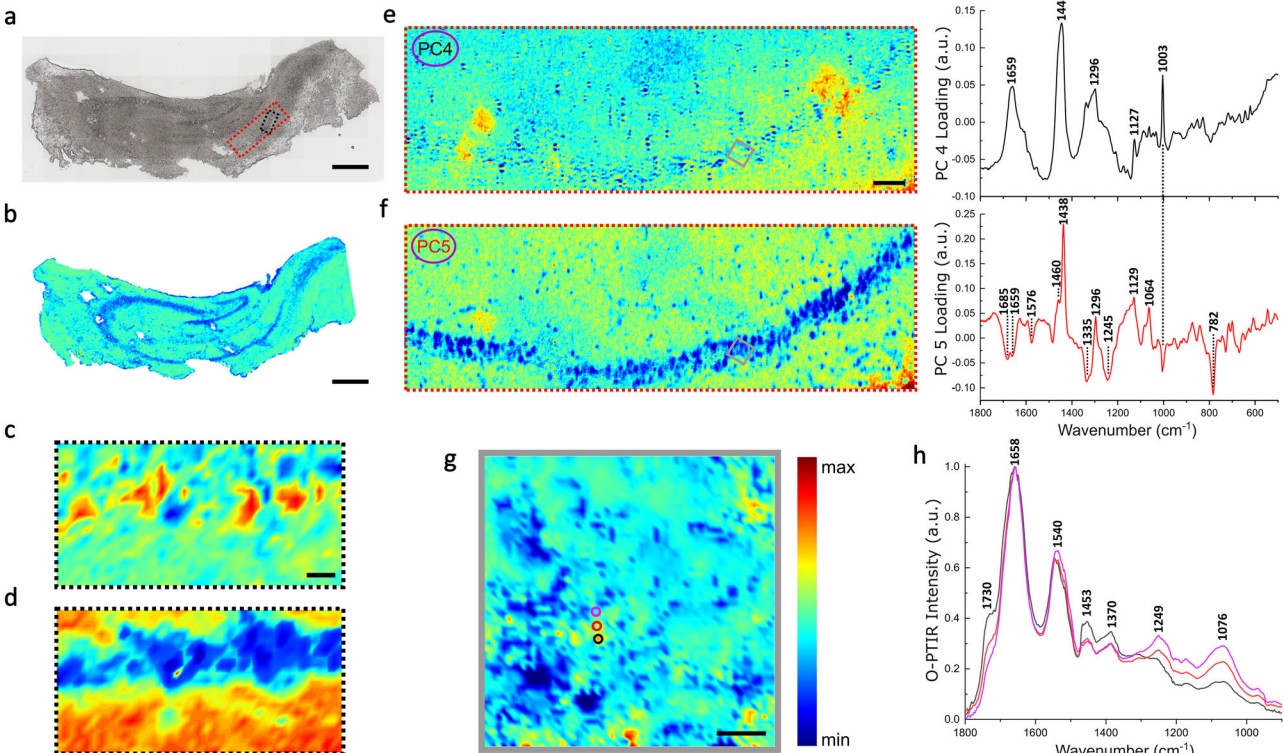

**Fig. 2 | Multimodal imaging of an ex vivo TG mouse hippocampal section conducted using FTIR spectroscopic imaging, SR-IR, Raman, and O-PTIR microscopy. a** White light image of the hippocampal section. The red and black dashed boxes indicate areas where hyperspectral Raman and SR-IR maps were obtained, respectively. Scale bar: 500 μm. **b** Micro-FTIR spectroscopic image based on the integrated absorbance of the lipid ester band (1761–1720 cm$^{-1}$). The blue arc denotes the pyramidal layer. Scale bar: 500 μm. **c, d** SR-IR maps based on the integrated absorbance of the Amide I (1716–1600 cm$^{-1}$) and lipid ester band (1761–1720 cm$^{-1}$). Scale bar: 20 μm. **e, f** Raman score maps (left panel) and loading plots (right panel) of principal components PC4 and PC5. PC4 loadings mainly contain signatures of proteins: 1659 (Amide I), 1446 cm$^{-1}$ ($CH_2$ bending), 1296 cm$^{-1}$ ($CH_2$ deformation), 1127 cm$^{-1}$ ($C$–$N$ stretching), and 1003 cm$^{-1}$ ($C$–$C$ symmetric stretching), whilst PC5 presents lipid signals at 1438 cm$^{-1}$ and 1296 cm$^{-1}$ ($CH_2$ deformation), 1129 cm$^{-1}$ and 1064 cm$^{-1}$ ($C$–$C$ skeletal stretching) as well as negative peaks associated with proteins and DNA: 1659 cm$^{-1}$ (Amide I), 1245 cm$^{-1}$ (Amide III) and 782 cm$^{-1}$ (ring breathing of DNA). The PC5 score map directly reveals the lipid distribution in the tissue, which highlights the location of pyramidal neurons as lacking lipid content. The grey box in the score maps indicates an area measured with O-PTIR microscopy. Scale bar: 50 μm. **g** O-PTIR image based on the 1730 cm$^{-1}$-to-1658 cm$^{-1}$ intensity ratio, with locations of representative spectra indicated by circles. The colour bar indicates a gradient from low (blue) to high (red) intensity for all false-colour images. Scale bar: 10 μm. **h** Representative O-PTIR spectra, max-min normalised, from selected locations in **g**.

As an emerging technique with super-resolution IR imaging capabilities, O-PTIR microscopy enables a more detailed investigation of the variations of lipids and proteins at a subcellular level. Here we present an O-PTIR image (Fig. 2g) based on the 1730 cm$^{-1}$-to-1658 cm$^{-1}$ intensity ratio, which illustrates the distribution of lipids, specifically cholesterol esters, relative to proteins in the tissue. In Fig. 2h, three selected O-PTIR spectra are max-min normalised, with the most intense peaks associated with the phosphate group of nucleic acids and possibly hyperphosphorylated tau protein (asymmetric phosphate stretching: 1249 cm$^{-1}$, symmetric phosphate stretching: 1076 cm$^{-1}$), proteins (Amide I: 1658 cm$^{-1}$; Amide II: 1540 cm$^{-1}$; $C$–$O$ stretching: 1370 cm$^{-1}$) and lipids (ester $C=O$ stretching: 1730 cm$^{-1}$; $CH_2$ bending: 1453 cm$^{-1}$). The variations of intensity for these peaks reflect the inherent biochemical heterogeneity at specific locations, allowing for the identification of lipid-rich regions in the mapped area.

## SRS imaging

We further harnessed the sensitivity of SRS imaging in the lipid analysis of TG and WT samples in the CH stretching region. Figure 3 displays SRS images of an area in the CA1 region of TG and WT mouse hippocampus at 2844 cm$^{-1}$ ($CH_2$ symmetric stretching) and 2930 cm$^{-1}$ ($CH_3$ symmetric stretching) and corresponding merged images. Signals at 2844 cm$^{-1}$ (Fig. 3a, b) are essentially due to the presence of lipids, whilst those at 2930 cm$^{-1}$ (Fig. 3c, d) are indicative of both proteins and lipids. In the merged images (Fig. 3e, f), there are noticeable morphological differences between TG and WT mice specimens. Protein-rich (nuclei; green arrow) and lipid-rich bodies (lipid droplets (magenta arrow) and lipid filaments (cyan arrow)) are apparent in the mouse hippocampus. Cell nuclei in the WT tissue are regularly distributed in the two or three layers of pyramidal neurons with a few lipid droplets and filaments around; however, pyramidal neurons in the TG mouse are sparser and more disordered, surrounded by a large number of lipid droplets and filaments.

To achieve a more accurate characterization of protein- and lipid-rich structures in the tissues, multivariate analysis through SOM-PCA was applied to the SRS data. Figure 4a, b present the score images of PC1 and PC2 for a region of interest (ROI) in a TG sample, and Fig. 4c, d show the corresponding ones for a WT. PC1 scores reveal a similar structural makeup to that observed in the merged images (Fig. 3e, f), whereas PC2 scores distinctly highlight nuclei and lipid-rich structures within the tissues. As displayed in Fig. 4e, f, PC1 represents the mean spectrum of the tissues, while PC2 shows prominent $CH_2$ (saturated lipid) bands (2844 cm$^{-1}$: $CH_2$ symmetric stretching; 2875 cm$^{-1}$: $CH_2$ asymmetric stretching) and low $CH_3$ bands (2943 cm$^{-1}$: $CH_3$ symmetric stretching; 2970 cm$^{-1}$: $CH_3$ asymmetric stretching). The TG sample exhibits a higher 2844-to-2930 cm$^{-1}$ intensity ratio, indicating a larger lipid content compared to the WT sample. The SOM-PCA results for the other four ROIs in these samples, illustrated in Supplementary Fig. SI-1, demonstrate consistent features in PC1 and PC2. However, separate classification processes applied to the different ROIs

**Table 1 | Band assignment of proteins and lipids in biological tissues**

| Wavenumber (cm⁻¹) | Vibrational mode | Assignment | References |
|---|---|---|---|
| Raman bands | | | |
| 782,1576 | Ring breathing | DNA/Nucleic acid | 32,51,52 |
| 1003 | $C–C$ symmetric stretching | Phenylalanine | 32 |
| 1064,1129 | $C–C$ skeletal stretching | Lipids | 53,54 |
| 1127 | $C–N$ stretching | Proteins | 52,55 |
| 1245 | Amide III | Proteins | 56 |
| 1296 | $CH_2$ deformation | Fatty acids | 57 |
| 1335 | $CH_3CH_2$ wagging | Collagen/Nucleic acid | 58 |
| 1438,1460 | $CH_2$ deformation | Lipids | 53,59,60 |
| 1446 | $CH_2$ bending | Proteins/Lipids | 53,54 |
| 1659 | Amide I | Proteins | 61,62 |
| 1685 | Amide I (disordered structure) | Proteins | 56 |
| 2844 | $CH_2$ symmetric stretching | Lipids | 60 |
| 2875 | $CH_2$ asymmetric stretching | Lipids | 60 |
| 2930 | $CH_3$ symmetric stretching | Proteins | 63 |
| 2943 | Chain end $CH_3$ symmetric stretching | Proteins | 59 |
| 2970 | $CH_3$ asymmetric stretching | Cholesterol ester | 59 |
| 3010 | $C=H$ stretching | Unsaturated lipids | 64 |
| IR bands | | | |
| 1076 | Symmetric phosphate stretching | Nucleic acid/ Phosphorylated tau proteins | 65,66 |
| 1249 | Asymmetric phosphate stretching | Nucleic acid/ Phosphorylated tau proteins | 67 |
| 1370 | $C–O$ stretching | Proteins | 67 |
| 1455 | $CH_3$ bending | Proteins | 51 |
| 1540 | Amide II | Proteins | 68 |
| 1658 | Amide I | Proteins | 67 |
| 1730 | Ester $C=O$ stretching | Lipids | 69 |

hinder the quantitative comparison of biochemical information between TG and WT mice. To address this challenge, we applied common k-means cluster analysis to evaluate the hyperspectral SRS data from all ROIs between TG and WT samples (10 areas in total, 5 TG and 5 WT), enabling a classification of biochemical features based on similarities between the spectra.

Merged images (Fig. 5a, c) at 2844 and 2930 cm⁻¹ of an ROI in TG and WT mice brain are presented along with the multivariate analysis results (Fig. 5b, d). It is possible to recognise anatomical features of the pyramidal layer, cell bodies, lipid droplets and filaments within this area. Supplementary Fig. SI-2 illustrates four additional ROIs in these specimens; white dashed boxes in the merged images of the whole hippocampus indicate the areas (#1–#4) where hyperspectral SRS imaging was performed. Thus, the whole set of spectra is assigned to four clusters that denote different biochemical features (Fig. 5b, d) with corresponding colour-coded cluster

centroids (Fig. 5e). It is apparent from these images that TG and WT tissues present a different morphology in terms of density and shape of the neuronal bodies. In particular, the TG sample has fewer cells with irregular shape/size and distinct branched lipid-rich structures and lipid droplets. Cluster A presents substantial $CH_2$ stretching intensity at 2844 cm⁻¹ and 2875 cm⁻¹, compared to the $CH_3$ stretching mode (2930 cm⁻¹), indicating a spectral profile with the highest lipid signal amongst the cluster centroids. In contrast, cluster D exhibits the opposite features in these bands with the lowest lipid signal. In Fig. 5b, d, cluster A (blue) denotes the lipid-dense regions in the tissue, while cluster D (yellow) denotes the regions with the lowest lipid content, i.e. the pyramidal neural bodies including nucleoli and cytoplasm. Clusters B and C represent the rest of the ROI. In the TG sample, a large proportion of spectra were categorised into the group with the second higher lipid signal - cluster B (cyan). Instead, cluster C (burgundy) is more prevalent in the WT sample. A statistical analysis of the four clusters across three pairs of TG and WT samples is presented in Fig. 5f. We notice a larger prevalence of lipid content in TG mice than in WT mice, evident from the fractions of clusters (A, B) with higher lipid signals (ca. 80% vs 60%). This is a significant difference which reflects the biochemical composition of the tissues and how this is affected by the pathology.

**Immunofluorescence imaging**

To unravel the pathological origin of elevated lipid content in the TG mouse hippocampus, we performed correlative fluorescence and SRS imaging on the immunochemically stained TG and WT mouse hippocampal tissues. Separate staining was carried out on the mouse hippocampal sections in order to minimise the interference from the labelling with antibodies for astrocytes (GFAP) and microglia (Iba-1). Moreover, nuclei were labelled with DAPI for the ease of identification of pyramidal neurons. Fluorescence images of TG and WT tissue sections stained with Iba-1 and GFAP are shown in Fig. 6a, b, as well as in Supplementary Figs. SI-3 and SI-6(a,b). It can be seen that a substantial number of activated microglia and astrocytes are detected in the TG mouse hippocampus, whereas very limited glial processes are present in the WT samples. A statistical analysis of the relative proportion of activated microglia and astrocytes is presented in Supplementary Fig. SI-4. The spatial segmentation and centroid spectra derived from k-means cluster analysis of SRS hyperspectral data of TG samples are shown in Fig. 6c, d and Supplementary Fig. SI-5, respectively. The corresponding results for WT samples are displayed in Supplementary Fig. SI-6. The clusters mainly exhibit a similar biochemical composition as those observed in the common k-means cluster analysis (Fig. 5). Cluster A represents the most lipid-dense regions, while cluster D still corresponds to nuclei gathering along the pyramidal layer within the tissue.

Interestingly, by comparing the fluorescence and SRS images in Fig. 6e, f, we observe some degree of overlap between activated microglia cell bodies and cluster A which exhibits the highest lipid content. Conversely, there is a lack of overlap between the regions identified as activated astrocytes and cluster A. These results indicate that in TG samples, the elevated lipid (cluster A) is plausibly a pro-inflammatory response of activated microglia involved in tau pathology.

**Discussion**

Alzheimer's disease has two main hallmarks in amyloidopathy and tauopathy as well as a range of dysfunctions in the brain. There has been a growing focus on tauopathy in recent years, owing to the challenges encountered in clinical trials targeting amyloid-beta[29,30]. While tau is increasingly recognised as a promising therapeutic target, the understanding of the biochemical changes in the brain with AD-tauopathy remains considerably limited. rTg4510 mouse is a widely used model for the study of tauopathy, and the progression of tau pathology has been well characterised in this model[31,32]. In this study, we used a range of vibrational microspectroscopic techniques to visualise the morphology and biochemistry of the hippocampal CA1 region in the brain of TG (rTg4510) and WT mice of 12 months of age (advanced stage of disease). The experimental approach

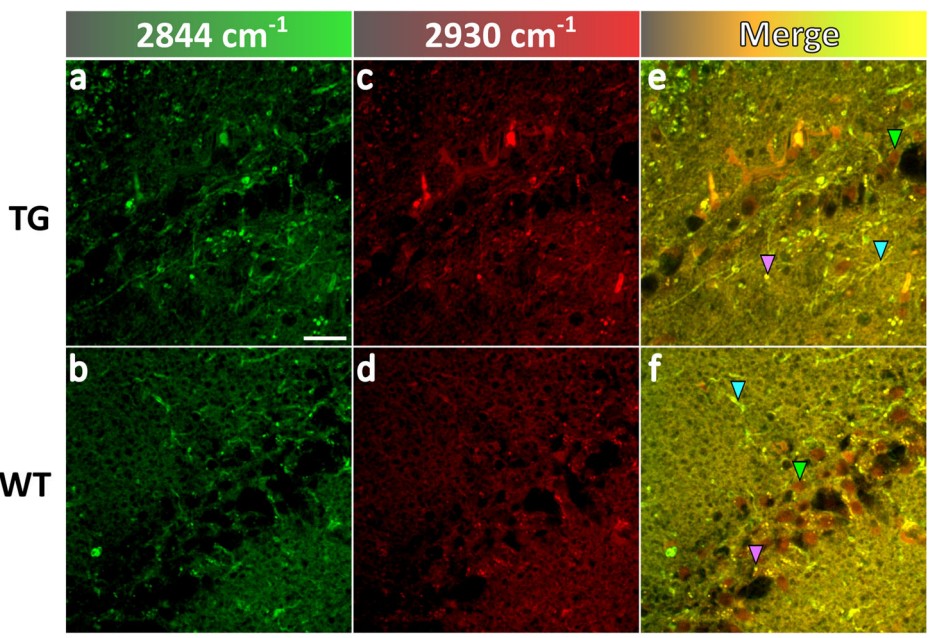

**Fig. 3 | Characterization of TG and WT mouse hippocampal sections with SRS imaging. a–d** SRS images of TG (top panel) and WT (bottom panel) pyramidal neurons and surrounding tissue acquired at 2844 cm⁻¹ ($CH_2$ symmetric stretching) and 2930 cm⁻¹ ($CH_3$ symmetric stretching). **e, f** Merged images at the two wavenumbers above for TG and WT samples with arrows indicating the nucleus (green), lipid droplet (magenta) and lipid-rich filament (cyan). Scale bar: 20 μm.

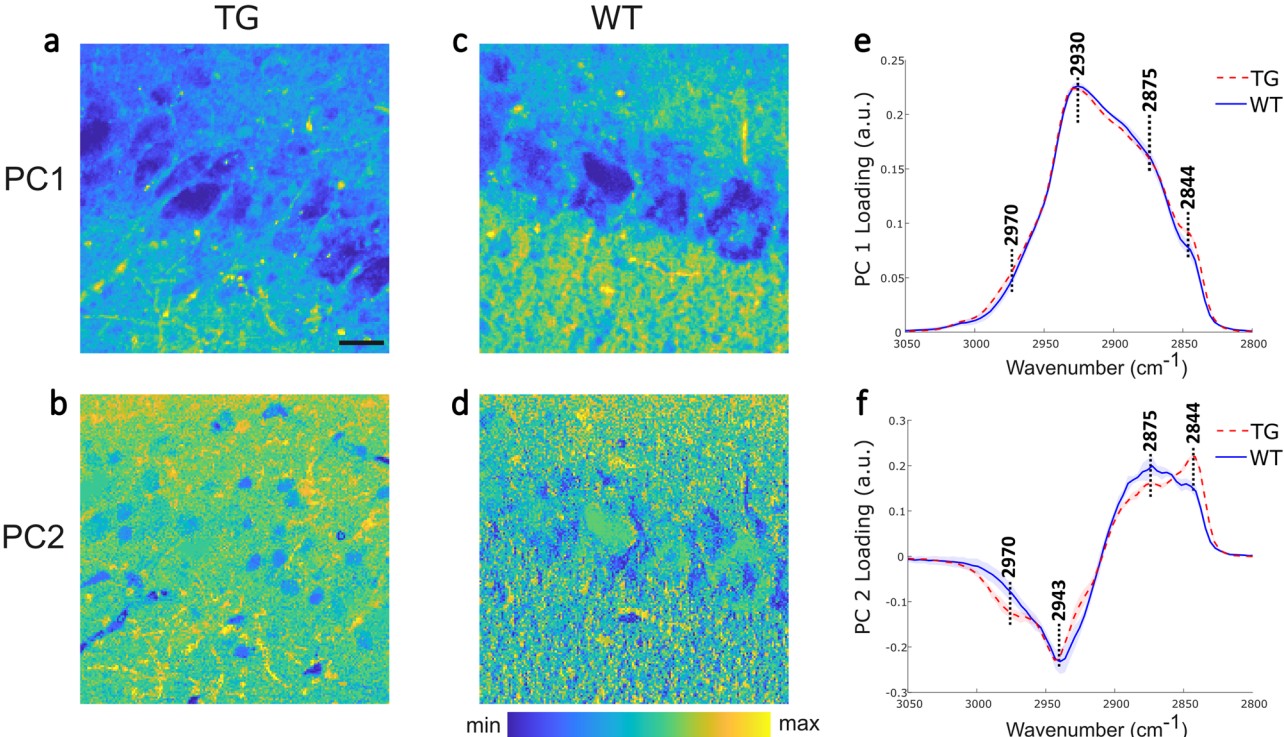

**Fig. 4 | Results of SOM-PCA applied to SRS hyperspectral maps of TG and WT mouse hippocampal sections. a–d** SOM-PCA score maps of PC1 and PC2 of an ROI in TG and WT samples. Scale bar: 20 μm; colour coding: blue (low) to yellow (high). **e, f** PC1 and PC2 loading plots (line) with standard deviation (shaded area) for the TG (red dashed) and WT (blue solid) samples. PC1 denotes the average spectrum of the tissue area, whilst PC2 represents the regions with high lipid-to-protein ratio. The peaks are at 2844 cm⁻¹ ($CH_2$ symmetric stretching), 2875 cm⁻¹ ($CH_2$ asymmetric stretching), 2943 cm⁻¹ ($CH_3$ symmetric stretching), and 2970 cm⁻¹ ($CH_3$ asymmetric stretching of cholesterol ester).

followed the order of increasing spatial resolution while decreasing field of view, from whole tissue to subcellular imaging. The capability of integrating multiple infrared and Raman methods in label-free chemical imaging of biological tissues is highly valuable in elucidating the structure and molecular make-up with subcellular resolution. To the best of our knowledge, it is the first time that a comprehensive array of spectroscopic techniques,

including the most recent super-resolution IR technique O-PTIR, were employed in an ex vivo study of tauopathy.

Whole tissue imaging (FTIR) enabled discrimination of the pyramidal layer, and higher resolution techniques (SR-IR, Raman, SRS and O-PTIR) were deployed on specific areas of the CA1 region in combination with multivariate statistical analysis. Common k-means cluster analysis enables

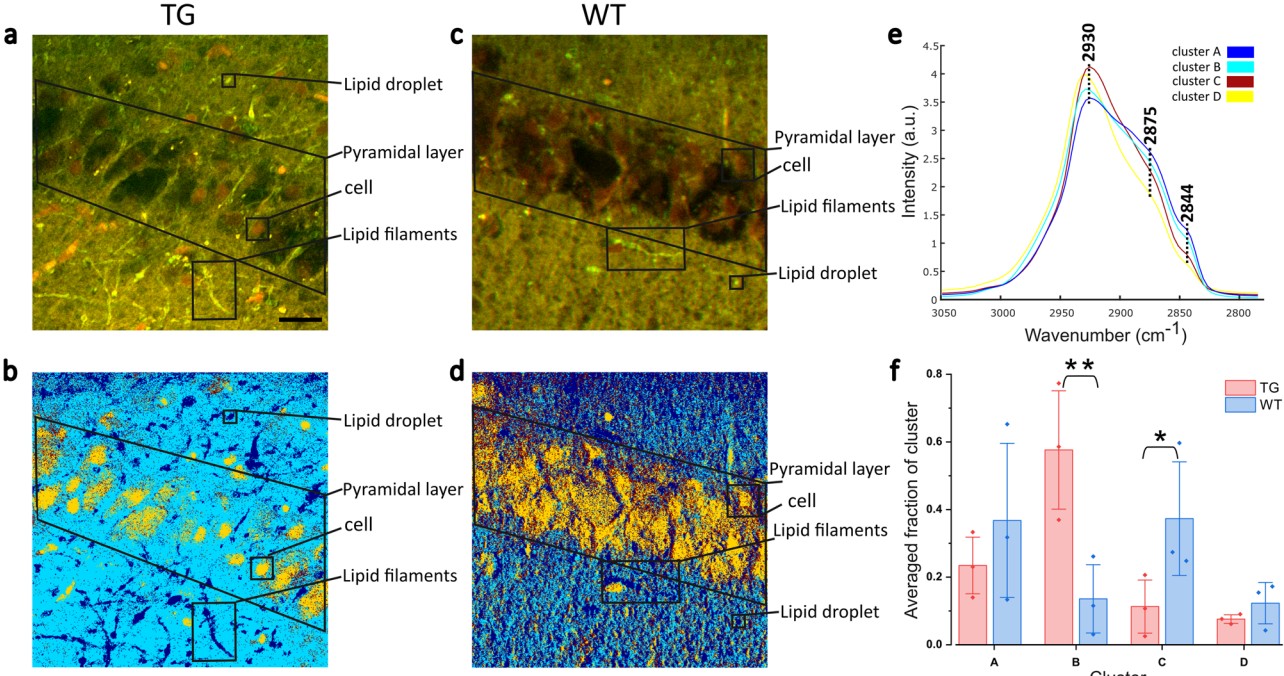

**Fig. 5 | Results of common k-means cluster analysis of SRS hyperspectral maps of TG and WT mouse hippocampal sections across an individual ROI (132.6 × 132.6 μm², 512 × 512 pixels). a, c** Merged images of TG and WT mouse hippocampus at 2844 cm⁻¹ and 2930 cm⁻¹. **b, d** Images derived from common k-means cluster analysis with four clusters. **e** Cluster centroid spectra. Main peaks are: 2844 cm⁻¹ ($CH_2$ symmetric stretching), 2875 cm⁻¹ ($CH_2$ asymmetric stretching) and 2930 cm⁻¹ ($CH_3$ symmetric stretching). **f** Bar plot showing the fraction of each cluster from three data sets of TG (red) and WT (blue) samples, including thirty ROIs in total. *$P < 0.1$, **$P < 0.01$. Scale bar: 20 μm (black).

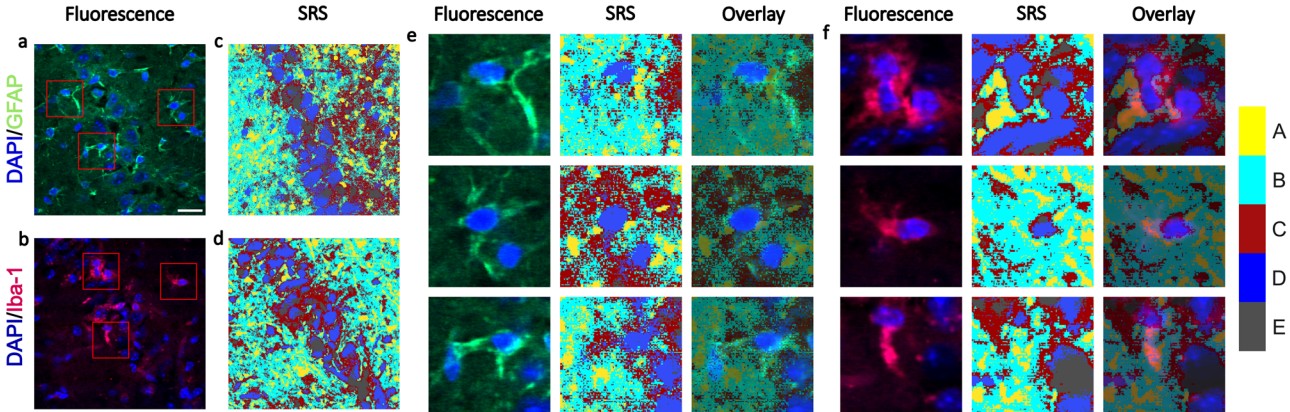

**Fig. 6 | Images derived from correlative immunofluorescence and SRS microscopy on TG tissue sections co-stained with DAPI (nuclei; blue) and either GFAP (astrocytes; green) or Iba-1 (microglia; magenta). a, b** Immunofluorescence images show the distribution of neurons (DAPI), astrocytes (GFAP) and microglia (Iba-1). Red boxes denote areas for comparison between the two techniques. **c, d** Spatial segmentation obtained from k-means cluster analysis with 5 clusters. **e, f** Close-up view of the fluorescence (left), k-means cluster analysis (middle) and overlay images (right). The partial correspondence between staining and clusters may be due to limitations in the staining protocol (i.e. without Triton X-100). Scale bars: 20 μm.

the direct comparison of biochemical composition across TG and WT mice. We observed a different morphology of the tissue as well as significant differences in lipid content between TG and WT samples, particularly a larger prevalence of lipid-rich structures in the TG sample. This result is expected based on the high sensitivity of our techniques to lipids and is in line with the finding of the collapse of neurons and disruption of lipid metabolism through AD progression[33].

To uncover the origin of the elevated lipid signal in TG tissues, correlative immunofluorescence and SRS imaging were conducted. It is noteworthy that fluorescence staining has no marked effect on the signal measured in SRS[34], owing to the low concentration of labelling agents and repeated washing during the staining process. Our findings show that some of the lipid deposits overlap with activated microglia in the TG mouse hippocampus, whereas WT samples show signals associated with a lower level of activation of microglia. This result aligns with previous studies showing that lipid droplets (LDs) are formed within microglia as a response to oxidative stress caused by ageing and neuroinflammation[4,5].

Tauopathy is known to be associated with metabolic changes, particularly lipid metabolic dysfunction. Previous studies have shown that lipid homeostasis is disrupted in AD brains with tau pathology, likely due to the

impaired glucose metabolism in tau-burdened cells[35,36]. As a response to metabolic stress, LDs are thought to be formed in neurons initially as a protective mechanism, followed by exacerbation of oxidative stress, contributing to neuronal damage and neuroinflammation. This will promote hyperactivity between neurons and glial cells, leading to an increased transfer of oxidised lipids to glial cells[37]. Our results expand upon this finding by demonstrating its relevance within the AD mouse model of tauopathy rTg4510 and highlighting that microglial LD burden contributes to neuroinflammation and neurotoxicity in tauopathy brains.

There have been controversial discussions about types of glial cells, astrocytes and microglia, that are associated with elevated lipids in brains[4,5,38]. Our previous work has demonstrated the presence of activated astroglia surrounding dense-core plaques in a mouse model of amyloidopathy[7]. Recently, lipid-droplet accumulating microglia have emerged as a new potential biomarker for neurodegenerative diseases including AD; the accumulation of LDs is also believed to impair the microglial tau phagocytosis and exacerbate tau pathology[4,8]. In particular, this phenomenon is thought to take place following the reduction in activation of AMP-activated protein kinase which has a critical role in regulating the homeostasis of brain lipids[39].

Our work further shows the importance of the involvement of the alteration of lipid metabolism and transport in neurodegeneration, specifically in tauopathy. Several research works have shown the causative relationship between dyslipidaemia, altered lipid transport across the blood brain barrier, altered lipid metabolism within the brain, and an increased risk of neurodegeneration-dependent dementia onset[40–43]. It is of pivotal importance to identify novel targets involved in the pathogenesis of neurodegenerative disorders. For example, recent phase III pre-clinical trials have shown the safety and efficacy of drugs such as Lecanemab and Donanemab as the first disease-course modifying agents ever developed for the treatment of Alzheimer's-related dementia[44]. However, these are characterised by high costs and limited efficacy[45]. For this reason, we believe that more attention on the lipid hypothesis for the pathogenesis of neurodegenerative diseases is granted and our observations reported in this work, together with the previous evidence of altered lipid signals in models of amyloid pathology[7,11,46], reinforce the need for further research in this area.

In conclusion, we demonstrated the use of a suite of vibrational microspectroscopic techniques, such as FTIR spectroscopic imaging, SR-IR, Raman, SRS and O-PTIR microscopy, for the label-free investigation of ex vivo brain tissue from a TG mouse model of Alzheimer's disease-associated tauopathy. The results revealed a strikingly distinct morphology and biochemistry between TG and WT mice. We observed a remarkable difference in lipid levels within the CA1 region of the hippocampus. Specifically, the TG mouse tissues displayed a significant presence of lipids compared with the WT samples. These lipid deposits correlate to some degree with activated microglia but not with astrocytes, which reconfirms the role of microglia in lipid regulation and dysregulation in pathophysiology. This finding adds weight to the hypothesis that lipid metabolism and its dysfunction is involved in AD-related tauopathy. Altogether our work provides new insight into the capabilities of label-free chemical imaging to detect and quantify complex morphological and biochemical variations of the brain affected by dementia-inducing tauopathy while reconfirming the evidence of microglia involvement in lipid regulation in AD pathology. Our findings can inform the development of novel, low-cost diagnostic tools for the early diagnosis of tauopathy-associated neurodegenerative disorders based on label-free chemical imaging.

## Methods
### Animal procedures
The animal procedures used in this work followed the UK Home Office Guidelines and the University of Exeter Animal Welfare Ethical Review Board. 12-month-old transgenic (TG) mice (rTg4510) and age-matched wild-type littermates (WT) were used in this study. This transgenic mouse model overexpresses human tau and maturely develops age-related NFTs across the hippocampus after the age of 5.5 months[31]. All animals were housed at room temperature under a 12-h light cycle, with access to food and water *ad libitum* before being euthanised.

### Tissue collection and sectioning
After the mice were sacrificed, their brains were immediately removed and cut into acute horizontal slices of 300 μm thickness with a vibratome, and then all slices were suspended in artificial cerebrospinal fluid, as previously described[47–49]. A number of samples of the hippocampal slices were retained for this study. After being post-fixed overnight with 4% formalin +0.1 M phosphate buffer solution (PBS), the slices were rinsed twice (5 min each time) and stored in 0.1 M PBS at around 4 °C. Then the slices were removed from the fridge, mounted between two glass coverslips and sealed with nail varnish, ready to be used for SRS imaging. Other slices from the same animals were immersed in 30% (w/v) sucrose solution for 24 h, embedded in optimal cutting temperature (OCT) medium, snap frozen, cryosectioned to 20 μm thickness, and placed onto Superfrost Plus microscope slides (Thermo Fisher) for immunohistochemical staining and fluorescence imaging. A few 300 μm thick slices from TG and WT mice were prepared using the same procedure above, cryosectioned to 20 μm thickness, mounted on Raman-grade polished calcium fluoride slides (Crystran, Poole, Dorset, UK) and dried at room temperature. A total of 67 samples from six rTg4510 and five WT mice were analysed using FTIR spectroscopic imaging, SR-IR, Raman, and O-PTIR microscopy.

### Microscopy and imaging
**Fourier transform infrared spectroscopic imaging.** A Fourier transform infrared (FTIR) spectroscopic imaging system, comprising of an Agilent Cary 670 FTIR spectrometer, a Cary 620 FTIR microscope with a 15x Cassegrain objective (NA = 0.62) and a liquid-nitrogen cooled focal plane array (FPA) detector (128 × 128 pixels), was used to acquire micro-transmission FTIR images. Resolutions Pro v. 5.3 software was used for acquisition of the data, whilst Matlab (R2021b) was used for data analysis. An infrared absorption spectrum was acquired for each pixel of the FPA detector by co-adding 32 interferograms and applying a Fourier transform. Spectra were obtained over the range 4000 to 1000 cm$^{-1}$ at a spectral resolution of 4 cm$^{-1}$ and a zero-filling factor of 2, allowing for a spectral spacing of 2 cm$^{-1}$. Before each measurement, a background was obtained in the absence of a sample by co-adding 64 interferograms. A whole tissue image was obtained as a mosaic of single tiles stitched together. Each tile was configured to be 128 × 128 pixels, corresponding to an area of 704 × 704 μm$^2$ on the specimen.

**Synchrotron radiation-infrared microscopy.** The microspectroscopy end station at MIRIAM beamline (B22) of Diamond Light Source was used for high spatial-resolution infrared hyperspectral mapping in FTIR mode. High brightness synchrotron infrared radiation (SR-IR) was coupled to a (Bruker UK) Vertex 80 V FTIR spectrometer and focused through the sample in a (Bruker UK) Hyperion 3000 Microscope equipped with 36x Cassegrain objective and condenser (NA = 0.5). The software Opus 8.5 was used for data acquisition and atmospheric correction ($CO_2$ and $H_2O$ gas and vapour removal, respectively). The transmitted light was collected by a high sensitivity 50 μm chip mid-band MCT detector with a cut-off of 650 cm$^{-1}$ (Infrared Associates). The sampled area for each spectrum was defined by an aperture to be 10 × 10 μm$^2$ at the sample plane, and the tissues were mapped at a step size of 5 μm (oversampling). 128 scans were co-added per spectrum in the range 4000–800 cm$^{-1}$ at 4 cm$^{-1}$ spectral resolution (zero filling factor 2). Absorbance spectra were calculated from sample transmission spectra using a reference i.e. background transmission spectrum (256 scans) recorded in a position off the tissue section but through the same calcium fluoride substrate.

**Optical photothermal infrared microscopy.** O-PTIR data were collected using a mIRage infrared microscope system (Photothermal Spectroscopy Corp.) equipped with a Cassegrain reflective objective (40×,

0.78 NA, 8.3 mm working distance) and a high-precision motorised stage. The mIRage system consists of a pump-probe setup where the pump is a tunable pulsed IR quantum cascade laser (QCL) and the probe is a continuous wave (CW) 785 nm laser. The QCL laser covers four wavenumber regions within the range of 1800–780 cm$^{-1}$ with 40–500 ns pulse duration and up to 100 kHz repetition rate. The spectra were acquired with 43% IR power and 35% probe power at a spectral resolution of ca. 2 cm$^{-1}$, an average of 9 scans and a scanning step size of 0.5 μm. Prior to all measurements, a pre-calibrated background was measured using a Kevley low-E substrate. The control of the mIRage system and background extraction were performed using PTIR Studio software, while data manipulation was done with Matlab (R2021b). The O-PTIR spectra, whilst being devoid of artifacts that are common in FTIR spectra, i.e. due to Mie scattering, have better specificity that enables features in sample materials to be detected on a smaller spatial scale[20].

**Raman microscopy.** Raman micro-spectroscopy maps were recorded using a Renishaw inVia Raman microscope equipped with a 785 nm diode laser, a Leica long working distance 50x (NA = 0.50) objective, and an x-y-z motorised stage. The backscattered light from the sample was collected by the same objective lens, sent to a 600 lines/mm diffraction grating and detected by a deep-depletion CCD camera. The measurements were controlled via WiRE v. 4.0 software. Raman maps were acquired in streamline mode using a laser line focus with an exposure time of 150 s, step size of 1.4 μm, and a spectral range of 2400–450 cm$^{-1}$. Spectral signals within the fingerprint region (1800–500 cm$^{-1}$) were retained for analysis.

**Stimulated Raman scattering microscopy.** A total of six specimens (3 TG and 3 WT) were analysed; these formed three pairs of samples. For each specimen, five ROIs were selected across the pyramidal layer for adequate sampling of the CA1 region, giving a total of thirty maps for statistical analysis. Stimulated Raman scattering was implemented using a spectral-focusing approach, previously described[50], which allows the Raman shift of interest to be rapidly tuned by controlling the time delay between chirped 120 fs pump and Stokes pulses. Synchronised, dual-wavelength ultrafast (120 fs) laser excitation was provided by an InsightX3 (Newport SpectraPhysics) which produces a fixed wavelength beam at 1041 nm and a tuneable beam between 680–1300 nm. The 1041 nm beam is used as the Stokes beam, whilst the tuneable beam is set at 802 nm and used as the pump beam. The pulses were chirped to several picoseconds using a spectral-focus and timing recombination unit (SF-TRU) (Newport SpectraPhysics) containing a pair of volume phase holographic gratings in each beam path with adjustable distances between the gratings to provide continuous, independent, dispersion control of the pump and Stokes beams to enable hyperspectral imaging with spectral resolution of 5 cm$^{-1}$. The two beams are spatially and temporally overlapped in the SF-TRU and the Stokes beam amplitude modulated at 19.5 MHz. The beams are then coupled into a modified confocal scanning microscope (Olympus FV3000) equipped with a 1.2 NA water immersion objective (Olympus UPlanSApo 60x). The transmitted beams, collected via a high numerical aperture condenser, are filtered to block the Stokes beam (Chroma CARS 890210 and Edmund optics 950 nm short pass filter) and detected on a silicon photodiode (APE). A lock-in amplifier (APE) is then used to extract the SRS signal, at 19.5 MHz from the laser intensity, providing a Raman intensity value at each pixel. Large area maps of the hippocampus were taken with the energy difference between the two beams tuned to the $CH_2$ and $CH_3$ stretching vibrations. Then hyperspectral scans were taken of regions of interest in the pyramidal layer over a spectral range of 3128–2780 cm$^{-1}$. This was achieved by taking a series of 101 images as the temporal delay between the two beams was scanned. The laser powers at the sample were 18 mW for the 1045 nm Stokes beam and 9 mW for the pump beam. SRS images (512 ×512 pixels) were acquired with a 10 μs pixel dwell time and 0.259 μm pixel size. The images were taken at a depth of approximately 10 μm below the surface of the tissue sections to avoid any potential artifacts due to the sample sectioning. False colour selection, scale bar, and colour merging of the images were performed with ImageJ software. $CH_2$ and $CH_3$ stretching images were generated by summing 7 SRS frames in a range of approximately 20 cm$^{-1}$ around 2844 cm$^{-1}$ and 2930 cm$^{-1}$ and presented in green and red, respectively.

**Immunohistochemical staining**
Immunohistochemical staining was performed to label the astroglia (anti-GFAP), microglia (Iba-1), and nuclei (DAPI) in the hippocampus of the mouse brain. Re-sectioned 20 μm thick mouse hippocampal sections were first washed in 0.1 M PBS three times for 10 min to remove the remaining sucrose and incubated in blocking buffer (10% NGS, PBS) for 20 min, followed by staining with the primary antibody anti-GFAP (1:1000, ab7260, Abcam) or anti-Iba1 (1:50, ab178847, Abcam) overnight at 4°C. Then the sections were rinsed in 0.1 M PBS three times for 10 min and incubated with Alexa Fluor secondary antibody (1:200 (GFAP), 1:200 (Iba1), ThermoFisher Scientific) for 2 h at room temperature. Then another three 10-min washes were performed, the sections were left to dry, and finally mounted with DAPI mounting medium (ab104139, Abcam). Note that the use of detergent (Triton X-100) for increasing the cell membrane permeability was skipped in the staining process, to avoid the interference of lipids and detergent.

**Two-photon fluorescence microscopy.** Two-photon excitation fluorescence imaging was performed in back-scattered (epi) geometry using the same microscope employed for the SRS imaging. The 1045 nm beam was shuttered and only the tuneable beam was used. In the SF-TRU box the diffraction grating was moved out of the beam path to enable fs imaging. For DAPI imaging the laser was tuned to 780 nm and for GFAP imaging the beam was tuned to 976 nm. The fluorescence signal was separated from the laser emission using a 775 nm short pass dichroic mirror (Chroma ZT 775sp-2p) and a fluorescence band pass filter 510 nm/80 nm FWHM (Chroma ET510/80) and then detected using a photomultiplier tube (Hamamatsu R3896). The imaging depth for the TPF images was set to match that of the corresponding SRS images.

**Data processing**
For FTIR imaging data, the absorbance of each band was calculated with a linear baseline drawn between the minima either side of the peak and absorbance values above this line were integrated with respect to wavenumber and plotted across the whole mosaic image area. The lipid ester signal ($C=O$ stretching) in the range 1720–1760 cm$^{-1}$ was used for the analysis. This enabled identification of the pyramidal layer (by low absorbance values; blue in Fig. 2b). SR-IR microscopy data were also subject to univariate analysis to obtain maps of Amide I (peptide $C=O$ stretching at ca. 1650 cm$^{-1}$) and lipid ester (1740 cm$^{-1}$). Raman microscopy data were subject to cosmic ray removal and standard normal variate normalization, and then analysed using principal component analysis (PCA). Ten PCs were computed, and selected PCs are presented here.

SRS datasets were analysed using a combined self-organising map - principal component analysis (SOM-PCA) algorithm[7]. SOM is an unsupervised machine-learning method normally used for dimensionality reduction and clustering of complicated data, which reshapes high-dimensional data into a low-dimensional network structure with weighting calculated from the input data. The combination of SOM and PCA allows for better performance in extracting and highlighting the lipid and protein distributions in the tissues. This combined method was applied to each ROI in both TG and WT samples. Ten PCs were obtained and the first two PCs were selectively shown. Both PCA and SOM-PCA were tested on Raman and SRS data, with only slight variations in the contrast of PCA scores, primarily due to differences in data manipulation. SRS datasets were also processed using k-means cluster analysis, which categorises hyperspectral data into clusters based on their spectral variances. Common k-means cluster analysis with 4 clusters was performed on each pair of TG and WT

samples, which enables a comparative study of chemical-specific features between the two groups of samples. An average fraction of the number of pixels that belong to each cluster was calculated based on all sample pairs, and Student t-tests were performed.

To enable a comparison with fluorescence imaging, SRS data were also analysed using common k-means cluster analysis with five clusters to segment each ROI. An additional cluster was included in the analysis to account for holes in the tissue sections that are filled with mounting medium (ab104139, Abcam). All the data manipulation was conducted using Matlab 2021b.

## Statistics and reproducibility
Information about statistical analyses, number of experimental replicates, and $p$ values are specified within the main text or in the Methods section. Data are shown as mean ± standard deviation (SD). Statistical analyses were performed with Origin 2019. The data presented were reproducible. In this study, a total of 67 samples from 6 transgenic and 5 wild type mice were analysed. For SRS measurements, 6 specimens (3 TG and 3 WT) with 5 ROIs each were selected to provide sufficient detection of morphological and biochemical signatures.

## Reporting summary
Further information on research design is available in the Nature Portfolio Reporting Summary linked to this article.

## Data availability
The authors declare that the data supporting the findings of this study are available within the paper, its supplementary information file, and on Figshare (https://doi.org/10.6084/m9.figshare.27144009.v1; https://doi.org/10.6084/m9.figshare.27143829.v1).

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

## Acknowledgements

This work was supported by the Wellcome Trust Institutional Strategic Support Award (WT105618MA), EPSRC CONTRAST Facility (EP/S009957/1), and an Alzheimer's Society Fellowship (to F.T.; ASJF-14-007). H.M. was supported by a China Scholarship Council/University of Exeter PhD scholarship. Diamond Light Source is acknowledged for beamtime SM27207 at the MIRIAM beamline B22. The animals were kindly provided by Eli Lilly Pharmaceuticals (Windlesham, UK) as part of larger collaboration with F.T. and the Alzheimer's Society. The authors wish to thank Dr Ryan Edginton (sample preparation and FTIR imaging data acquisition), Dr Pascaline Bouzy (O-PTIR measurements), Dr Gabriella Margetts-Smith and Meg Elley (immunohistochemical staining). For the purpose of open access, the author has applied a Creative Commons Attribution (CC BY) licence to any Author Accepted Manuscript version arising from this submission.

## Author contributions

F.P. and F.T. conceived, designed and supervised the project. F.T. provided the samples. H.M., A.E., J.Ma., M.B. and M.F. performed the data acquisition and data analysis. F.P., J.Ma., J.Mo., F.T., M.F., G.C. and N.S. helped with the study design and discussion of the results. H.M. wrote the manuscript with contributions from all other authors.

## Competing interests

The authors declare no competing interests.
