## [Transparent Peer Review file · Communications Biology]

Identification of tauopathy-associated lipid signatures in Alzheimer's disease mouse brain using label-free chemical imaging

Corresponding Author: Professor Francesca Palombo

Version 0:

Reviewer comments:

Reviewer #1

(Remarks to the Author)
Review Attached

Reviewer #2

(Remarks to the Author)

The work by Meng et. al., is a very nice study comparing lipid (and other biochemical) signatures in animal models of AD, using a variety of vibrational spectroscopic techniques (FTIR, SR-FTIR, OPT-IR, Raman, SRS). The direct chemical imaging has been complemented with immuno-fluorescence. From a methods point of view, this study has great value, allowing side by side comparison of a suite of spectroscopic methods, each offering advantages with respect to sensitivity, spatial resolution, and nature of chemical information revealed. Indeed, some of the sub-cellular resolution images shown in this paper are quite striking. The study also further confirms the role of disturbed lipid homeostasis in AD, and as such, adds further mechanistic value to the AD research field. I am more than happy to recommend publication following several minor amendments.

1. As this paper focusses on the role of lipids in AD, and specifically the lipid halos that surround amyloid plaques, there are two key papers that should be cited (below). Indeed the paper by Summers et al. presents a comparison of lipid imaging around plaques with FTIR, SR-FTIR, confocal Raman microscopy, and tissue autofluorescence.

Summers, Kelly L., Nicholas Fimognari, Ashley Hollings, Mitchell Kiernan, Virginie Lam, Rebecca J. Tidy, David Paterson et al. "A multimodal spectroscopic imaging method to characterize the metal and macromolecular content of proteinaceous aggregates ("amyloid plaques")." *Biochemistry* 56, no. 32 (2017): 4107-4116.

Benseny-Cases, Núria, Oxana Klementieva, Marine Cotte, Isidre Ferrer, and Josep Cladera. "Microspectroscopy (μ FTIR) reveals co-localization of lipid oxidation and amyloid plaques in human Alzheimer disease brains." *Analytical Chemistry* 86, no. 24 (2014): 12047-12054.

2. On Page 3 line 121. "As an emerging technique with super-resolution capabilities" The term "super resolution" isn't appropriate, given that term is generally associated with "super-resolution" microscopy, which has much better spatial resolution than OPT-IR. Perhaps "As an emerging technique with improved-resolution capabilities" or "As an emerging technique with resolution capabilities better than the diffraction limit of infrared light" would be more appropriate.

3. For the snap freezing of tissues, how was this performed (-20 C freezer, -80 C freezer? Dry Ice? Liq N2)? Further, I don't believe the scientific conclusions of this study have been impacted, so this comment can mostly be ignored, however, from a pure histology / microscopy point of view, the tissue preservation of the sample shown in Figure 2 A, is poor, with obvious ice-crystal damage occurring at the periphery of the tissue (i.e. "swiss cheese appearance"). Apologies for sounding critical, and as stated above, I don't think it has impacted the outcomes of the study, but it should be fairly straightforward to obtain tissue sections with much better preservation than that shown in Figure 2A.

Author Rebuttal letter:

Manuscript COMMSBIO-24-3021

We would like to thank the reviewers for their positive and constructive comments on our manuscript. Below we have addressed each comment and, when necessary, made amendments to the text highlighting the changes in the manuscript file. In the remainder, the reviewers' comments are in normal text, our response in italic, and changes applied to the manuscript in blue font.

Reviewer #1:

This work titled "Elucidating signatures of tauopathy in Alzheimer's disease mouse brain by label-free chemical imaging" by Meng et al. uses different modalities of spectroscopic imaging to characterize the chemical variation between diseased and normal brain tissues in mouse models. The manuscript is well written, and the results are interesting. However, there are some key aspects that need to be revised before this work can be recommended for publication, as detailed below.

1. Why are so many techniques necessary? The use of two complementary modalities, such as Raman and IR is understandable. However, why are multiple types of Raman and IR imaging necessary? For example, the spectral information obtainable from all three of the IR based techniques reported here is the same; so why use all three of them? If one of them is better suited for the investigation, why not use that? The same argument applies to using both Raman and SRS. The benefit of using so many approaches is unclear and needs to be described in detail in the manuscript. Without this explanation, it seems the authors are not sure what the suitable technique is for this work and the experimental approach seems exploratory at best.

We appreciate the reviewer's comment, and we note that the use of multiple imaging techniques, even with the same underlying principle, offers the ability to start from a large scale, whole-tissue imaging (e.g. FTIR) and, by increasing the resolution, move to a more detailed analysis of subcellular features (OPTIR). Although this may seem an overkill, we believe it is important to address the problem of tauopathy from a multiscale standpoint. In fact, the chemical composition of the brain is a complex problem in physiological conditions and further uncertainty is introduced by the appearance of chemical species which become progressively insoluble and which are causative of neuronal degeneration such as tauopathy. For this reason, it is important to use a suite of micro-spectroscopic methods, each offering advantages with respect to sensitivity, field of view, spatial resolution, imaging speed, and nature of chemical information revealed. FTIR, SR-IR and O-PTIR imaging were performed in the order of an increasing spatial resolution and subsequently a decreasing field of view. We were able to conduct whole tissue imaging with FTIR microscopy as a benefit of larger field of view and diffraction-limited resolution (to approx. 5 - 10 μm); however, it will be rather time-consuming to map the whole tissue with high-resolution O-PTIR imaging (resolution of approx. 0.6 μm with a 785 nm laser). For example, the O-PTIR imaging of an ROI of 20 \times 20 μm^2 with a 0.5 μm step size will normally take 6 hours. Therefore, here we used FTIR and SR-IR microscopy to show the variations of lipids and proteins across a large scale, while O-PTIR microscopy was used for imaging at subcellular level in specific ROIs. A similar principle has driven the choice of the complementary use of Raman and SRS microscopy. The imaging resolution of SRS is better than Raman, however Raman offers a more specific approach from the measurement of signals in the fingerprint region (500-1800 cm^{-1}) (compared with only the CH stretching region accessed by SRS). We believe that Figure 1 in the manuscript illustrates this point, however we have modified the caption which now reads "Visual illustration of our approach where a suite of micro-spectroscopic methods was applied following the order of increasing spatial resolution and decreasing field of view, from whole tissue to subcellular imaging" and added a sentence in the Discussion: "The experimental approach followed the order of increasing spatial resolution while decreasing field of view, from whole tissue to subcellular imaging."

2. It is somewhat surprising that the authors have not investigated protein aggregates and their association with lipids in this work. The pathological significance of extracellular and intracellular protein aggregates in AD and other tauopathies is well known, and their associations with lipids has been attracted significant interest in the past decade. The authors need to add a discuss in the manuscript to clearly delineate the reason behind their choice.

We agree with the reviewer that protein aggregates (in association with lipids) are an area of major focus in AD research. Indeed, our work aims at identifying and quantifying both protein and lipid signals. In addition, we have investigated the potential origin of the altered lipid signal in tau pathology by using immunohistochemical methods. To better

clarify the potential role of lipid-protein interactions in our work, we have now added this sentence to the manuscript (Discussion): "In particular, this phenomenon is thought to take place following the reduction in activation of AMP-activated protein kinase which has a critical role in regulating the homeostasis of brain lipids.³⁹"

3. The authors need to describe the significance of this work in more detail. The key insights obtained from this work and their importance in the context of neurodegenerative diseases is not very clear and can be better detailed.

We addressed this point by clarifying the potential clinical impact of our work in the Discussion: "Tauopathy is known to be associated with metabolic changes, particularly lipid metabolic dysfunction. Previous studies have shown that lipid homeostasis is disrupted in AD brains with tau pathology, likely due to the impaired glucose metabolism in tau-burdened cells.^{35,36} As a response to metabolic stress, LDs are formed in neurons initially as a protective mechanism, followed by exacerbation of oxidative stress, contributing to neuronal damage and neuroinflammation. This will promote hyperactivity between neurons and glial cells, leading to an increased transfer of oxidised lipids to glial cells.³⁷ Our results expand upon this finding by demonstrating its relevance within the AD mouse model of tauopathy rTg4510 and highlighting that microglial LD burden contributes to neuroinflammation and neurotoxicity in tauopathy brains."; "Our work further shows how important is the involvement of the alteration of lipid metabolism and transport in neurodegeneration, specifically in tauopathy. Several research works have shown the causative relationship between dyslipidaemia, altered lipid transport across the blood brain barrier, altered lipid metabolism within the brain, and an increased risk of neurodegeneration-dependent dementia onset.⁴⁰⁻⁴³ It is of pivotal importance to identify novel targets involved in the pathogenesis of neurodegenerative disorders."; "For this reason, we believe that more attention on the lipid hypothesis for the pathogenesis of neurodegenerative diseases is granted and our observations reported in this work, altogether with the previous evidence of altered lipid signals in models of amyloid pathology,^{7,11,46} reinforce the need for further research in this area."

4. Figure 5 : why does the blank region correspond to a cluster? This region should not contribute any spectral signal; so why is it recognized as a spectral class in the analysis?

This is a good point. In fact, we believe that the holes present in the tissue sections get filled with mounting medium. So the spectrum in Fig. S1-5 reflects this presence. To clarify this association, in the manuscript (Data processing) we have modified the relevant sentence which now reads "An additional cluster was included in the analysis to account for holes in the tissue sections that are filled with mounting medium (ab104139, Abcam)."

5. I would suggest the authors rephrase the title to be more specific about the exact findings of this work. The title seems to suggest the authors are looking at tau aggregates, but they are only looking at microglial activation

We agree with the reviewer and replaced the title with: "Identification of tauopathy-associated lipid signatures in Alzheimer's disease mouse brain using label-free chemical imaging".

6. The authors need to provide a statistical information related to data, such as number of ROIs etc. in the manuscript. Without this information, it is difficult to judge the significance of the findings reported.

This information was already provided (section 4.3.5), however we have now added a line to specify our choice: "A total of six specimens (3 TG and 3 WT) were analysed; these formed three pairs of samples. For each specimen, five ROIs were selected across the pyramidal layer for adequate sampling of the CA1 region, giving a total of thirty maps for statistical analysis."

7. The authors need to cite recent work by other groups that have used different modalities of IR, and Raman to investigate AD tissue specimens.

As also suggested by Reviewer 2, we have now added two references (Introduction [14,15]):

Summers, Kelly L., Nicholas Fimognari, Ashley Hollings, Mitchell Kiernan, Virginie Lam, Rebecca J. Tidy, David Paterson et al. "A multimodal spectroscopic imaging method to characterize the metal and macromolecular content of proteinaceous aggregates ("amyloid plaques")." *Biochemistry* 56, no. 32 (2017): 4107-4116.

Benseny-Cases, Núria, Oxana Klementieva, Marine Cotte, Isidre Ferrer, and Josep Cladera. "Microspectroscopy (μ FTIR) reveals co-localization of lipid oxidation and amyloid plaques in human Alzheimer disease brains." *Analytical Chemistry* 86, no. 24 (2014): 12047-12054.

Reviewer #2:

The work by Meng et. al., is a very nice study comparing lipid (and other biochemical) signatures in animal models of AD, using a variety of vibrational spectroscopic techniques (FTIR, SR-FTIR, OPT-IR, Raman, SRS). The direct chemical imaging has been complemented with immuno-fluorescence. From a methods point of view, this study has great value, allowing side by side comparison of a suite of spectroscopic methods, each offering advantages with respect to sensitivity, spatial resolution, and nature of chemical information revealed. Indeed, some of the sub-cellular resolution images shown in this paper are quite striking. The study also further confirms the role of disturbed lipid homeostasis in AD, and as such, adds further mechanistic value to the AD research field. I am more than happy to recommend publication following several minor amendments.

We truly appreciate the reviewer's positive feedback on our manuscript, especially recognising the great value of this study.

1. As this paper focusses on the role of lipids in AD, and specifically the lipid halos that surround amyloid plaques, there are two key papers that should be cited (below). Indeed the paper by Summers at all presents a comparison of lipid imaging around plaques with FTIR, SR-FTIR, confocal Raman microscopy, and tissue autofluorescence.

Summers, Kelly L., Nicholas Fimognari, Ashley Hollings, Mitchell Kiernan, Virginie Lam, Rebecca J. Tidy, David Paterson et al. "A multimodal spectroscopic imaging method to characterize the metal and macromolecular content of proteinaceous aggregates ("amyloid plaques")." *Biochemistry* 56, no. 32 (2017): 4107-4116.

Benseny-Cases, Núria, Oxana Klementieva, Marine Cotte, Isidre Ferrer, and Josep Cladera. "Microspectroscopy (μ FTIR) reveals co-localization of lipid oxidation and amyloid plaques in human Alzheimer disease brains." *Analytical Chemistry* 86, no. 24 (2014): 12047-12054.

We thank the reviewer for recommending these papers and agree that they fit well within the context of our work. So we have now added them to the manuscript as the new references 14 and 15 in the Introduction.

2. On Page 3 line 121. "As an emerging technique with super-resolution capabilities" The term "super resolution" isn't appropriate, given that term is generally associated with "super-resolution" microscopy, which has much better spatial resolution than OPT-IR. Perhaps "As an emerging technique with improved-resolution capabilities" or "As an emerging technique with resolution capabilities better than the diffraction limit of infrared light" would be more appropriate.

We note that in the context of pump-probe IR techniques, such as the O-PTIR microscopy used in this work, the word super-resolution is appropriate in that the diffraction limit of IR light is beaten using shorter wavelengths probe lasers. However, we understand that this might cause a confusion with more widely used super-resolution techniques such as STORM. For this reason, we have now specified in Section 2.1 "As an emerging technique with super-resolution IR imaging capabilities".

3. For the snap freezing of tissues, how was this performed (-20 C freezer, -80 C freezer? Dry Ice? Liq N₂)? Further, I don't believe the scientific conclusions of this study have been impacted, so this comment can mostly be ignored, however, from a pure histology / microscopy point of view, the tissue preservation of the sample shown in Figure 2 A, is poor, with obvious ice-crystal damage occurring at the periphery of the tissue (i.e. "swiss cheese appearance). Apologies for sounding critical, and as stated above, I don't think it has impacted the outcomes of the study, but it should be fairly straight forward to obtain tissue sections with much better preservation that that shown in Figure 2A.

We agree that the tissue sections may look worn. We would like to point out that this tissue has been used for multiple technique studies. It was originally freshly sliced for electrophysiology applications, subsequently post-fixed for histological measurements and finally snap frozen following cryoprotection with sucrose (30% in PBS). While we appreciate that the low quality of the cryosectioning could be an issue, we still see a clear signal while working within the framework of the 3 R's in animal research reducing the number of mice sacrificed for this study. In the Methods section, we stated: "Other slices from the same animals were immersed in 30% (w/v) sucrose solution for 24 hours, embedded in optimal cutting temperature (OCT) medium, snap frozen, cryosectioned to 20 μ m thickness".

Version 1:

Reviewer comments:

Reviewer #2

(Remarks to the Author)

I have read through the revised manuscript, in addition to all reviewer comments, and the authors response to reviewers, and I am satisfied with the additions/revisions. In my opinion, this manuscript is suitable for publication.

I do agree with reviewer 1 that it is a pity protein aggregates have not also been studied. However, there is substantial discussion in the field at the moment about whether or not protein aggregates do have a causal effect in neurodegeneration (perhaps they are a consequence of the disease instead), so I admit it is somewhat refreshing to read a paper not exclusively focussed on protein aggregation. In addition, in my opinion, the methodological advancements with respect to direct lipid imaging presented in this study, are sufficient to warrant publication.
